# Difference-aware Decision Learning for Multimodal Image Fusion

**Hao Pan**[1]  **Jian Dai**[2]  **Yuan Sun**[3]  **Zhenwen Ren**[*1]  **Xingfeng Li**[*1]

## Abstract

Multimodal image fusion aims to integrate complementary information from different modalities, but cross-modal discrepancies and local conflicts often make modality allocation uncertain, causing information loss or artifact propagation. We address this problem by formulating fusion as an observation-conditioned probabilistic decision-learning problem, where local modality contribution is explicitly modeled as a decision variable. Based on this view, we propose a d**I**fference-aware **D**ecision-l**EA**rning mu**L**timodal image fusion paradigm (**IDEAL**). IDEAL uses cross-modal differences as decision triggers and constructs spatial and spectral decision conditions from multi-scale difference attention, power-spectrum energy, complementary spectra, and spectral-entropy reliability. These conditions are mapped to interpretable contribution policies through a symmetric Beta prior, while uncertainty modulation pulls unreliable decisions toward conservative mixing when evidence is insufficient. Extensive experiments on multiple fusion tasks demonstrate stable and competitive performance against state-of-the-art methods. Code is available at: `https://github.com/Pon915/IDEAL-main`.

## 1. Introduction

Multimodal image fusion aims to integrate complementary information from different imaging conditions or sensing mechanisms, thereby improving the reliability of downstream perception and understanding (Liu et al., 2024; Li et al., 2026c; Luo et al., 2019; Peng et al., 2026; Li et al., 2024a; 2026b; Jie et al., 2025; Peng et al., 2024). For in-

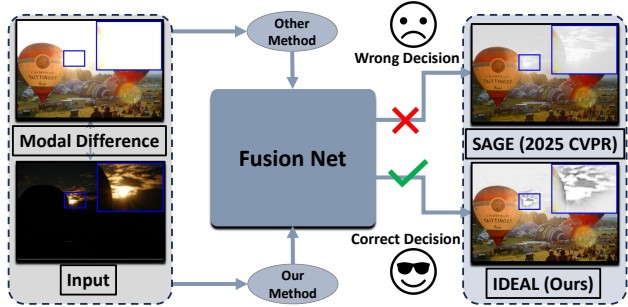

*Figure 1.* Qualitative examples of IDEAL and SAGE (Wu et al., 2025) on multi-exposure images. In regions with strong cross-modal discrepancies, baseline methods often make biased fusion decisions. Our method instead treats these differences as explicit observations, enabling more reliable decisions and higher-quality fusion.

stance, infrared–visible fusion exploits cross-spectrum cues to obtain richer multispectral image information (Yang et al., 2026; Li et al., 2025d; Chen et al., 2026), medical fusion combines structural and functional modalities to support clinical judgment (Li et al., 2025b), and multi-focus fusion merges sharp regions across focal planes to obtain an all-in-focus image (Pan et al., 2026).

Despite diverse applications (Wang et al., 2020; Li et al., 2025a; Wang et al., 2022; Li et al., 2024b; Shi et al., 2026; Li et al., 2026a), these problems share a common challenge, *i.e.*, when faced with the same spatial area, it is difficult to make reasonable and efficient decisions regarding the retention of different information. Therefore, fusion quality critically depends on the model's decision strategy *i.e.*, how to allocate modality contributions, preserve discriminative information, and suppress erroneous signals to avoid information loss and artifact propagation.

Traditional fusion works (Hill et al., 2016; Cai et al., 2017; Li et al., 2018) typically relied on classical mathematical tools for explicit modeling, thereby the fusion task was often formulated as a transform-domain information recomposition or an optimization problem. Researchers allocate contributions by manually designing rules. However, because these methods rely heavily on manually designed fusion rules and idealized assumptions, which make them struggle to effectively handle the complex situations encountered in real-world scenarios, resulting in notable limitations in

*Corresponding author [1]Southwest University of Science and Technology, Mianyang, China [2]Southwest Automation Research Institute, Mianyang, China [3]Sichuan University, Chengdu, China. Correspondence to: Zhenwen Ren <rzw@njust.edu.cn>, Xingfeng Li <lixingfeng@njust.edu.cn>.

*Proceedings of the 43rd International Conference on Machine Learning*, Seoul, South Korea. PMLR 306, 2026. Copyright 2026 by the author(s).

both flexibility and generalization. With the advent of deep learning, researchers have leveraged the powerful nonlinear fitting capacity of deep neural networks, substantially enhancing the performance of fusion models. The powerful nonlinear fitting capabilities of DL have propelled multimodal image fusion to new heights.

Recent representative works (Zhang et al., 2020; Li et al., 2025c; Tang et al., 2023b; Wang et al., 2025; Zhang et al., 2026; Su et al., 2025) primarily enhance deep fusion performance by scaling up network capacity to improve its nonlinear fitting ability. CDDFuse (Zhao et al., 2023b) and SwinFusion (Ma et al., 2022) instead expand the model's effective receptive field to mitigate the lack of global semantics caused by small receptive fields. EMMA (Zhao et al., 2024a) proposes a novel training paradigm that effectively provides stronger supervision under unsupervised losses, guiding the model more explicitly during optimization. Moreover, an increasing number of methods leverage the strong transferability of foundation models as an expert system, using text or other auxiliary inputs to modulate fusion features and obtain more powerful representations (Zhao et al., 2024b; 2023c; Yi et al., 2024). Alternatively, some works couple fusion with downstream tasks (*e.g.*, semantic segmentation and object detection) to encourage the fusion model to encode more task-relevant information (Tang et al., 2023a; Zhao et al., 2023a; Yang et al., 2025).

Although existing methods have achieved impressive performance, they still implement the fusion process as an implicit feature aggregation or modulation process. Attention maps, gating weights, and semantic priors introduced by large-scale models all, to varying degrees, determine the contribution allocation of each modality in a local region. However, these contribution allocations are usually generated in the form of deterministic activation or implicit feature transformations, and are not explicitly modeled as fusion decisions constrained by observation conditions. Therefore, when significant cross-modal differences appear in a local region, the model often struggles to determine the true meaning of these differences in the fusion decision: they may correspond to valid complementary evidence of one modality relative to another, or they may simply be interference caused by imaging degradation, strong response bias, or locally unreliable observations. Without explicit characterization of the nature of differences and the reliability of evidence, models can easily mistake stronger response for more reliable information, thus creating bias in contribution allocation. As shown in Figure 1, when a modality is overexposed in a local area, its pixel intensity will show a significantly high response. However, this response reflects more image degradation than effective information enhancement. In this case, existing methods often struggle to make reliable contribution judgments, easily assigning excessive weight to the overexposed mode, thereby suppressing the weaker but

valuable texture and structural information in another mode, ultimately resulting in the loss of local information.

Therefore, we formulate multimodal fusion as a modality contribution decision allocation process. Rather than directly producing deterministic contribution weights from unconditional feature activations, reliable fusion should first construct explicit observations from the inputs and then infer local contribution decisions conditioned on these observations. To this end, we propose IDEAL, a difference-aware decision learning framework. IDEAL converts cross-modal discrepancies into decision-related observations through two components: a modality difference attention module, which identifies regions requiring local contribution decisions, and a conditional decision gating module, which extracts frequency-domain observations from spectral energy, complementary spectra, and entropy-based reliability to assess reliable complementary evidence. These observations parameterize a symmetric Beta prior for local contribution decisions, where the prior mean serves as an interpretable gating policy and the evidence strength reflects decision uncertainty. When observations are reliable, IDEAL performs explicit modality assignment; when evidence is insufficient, uncertainty modulation pulls the policy toward a conservative hybrid state, preventing unreliable cues from dominating fusion. The main contributions of this work are summarized as follows:

- We reformulate multimodal image fusion as observation-conditioned probabilistic decision learning, where local modality contribution is modeled as an explicit decision variable. This provides a principled framework for decision-aware fusion under cross-modal discrepancy and uncertainty.

- We propose an explicit observation-to-decision paradigm that converts cross-modal discrepancy, complementary evidence, and reliability into decision-relevant observations. These observations induce interpretable priors over modality contribution, enabling controllable prior decision formation.

- We designed a novel decision module suitable for prior decision, which consists of a Modal Difference Attention (MDA) module and a Conditional Decision Gating (CDG) module.

- We conducted extensive experiments on general multimodal image fusion tasks, and the results show that our method outperforms state-of-the-art methods.

## 2. Formalization

We formalize multimodal image fusion as an observation-conditioned decision-learning problem. Different from conventional formulations that regard fusion as deterministic

feature aggregation, our formulation emphasizes that the fused image is controlled by local modality contribution decisions. This section focuses on two key points: (i) contribution decisions are particularly important in discrepancy or complementary regions; (ii) explicit observation-conditioned decision learning can produce more reliable contribution policies in such regions.

Let the multimodal input be a collection of $M$ modalities:

$$X = \{X^1, \ldots, X^M\}. \tag{1}$$

The fusion output lies in $Y \in \mathcal{Y}$. We introduce a decision variable $Z \in \mathcal{Z}$, which represents internal fusion controls such as attention, gating, modulation, and modality selection. Instead of directly modeling fusion as $Y = \mathcal{F}_\omega(X)$, we formulate it as a decision-conditioned mapping:

$$Y = \mathcal{F}_\omega(X, Z), \tag{2}$$

where $\omega$ parameterizes the fusion operator. This formulation makes explicit that the fusion result is determined not only by the source images $X$, but also by the local contribution decision $Z$.

## 2.1. Decision Sensitivity in Discrepancy Regions

For clarity, we consider the two-modality case $X = \{X_A, X_B\}$. For a local region $r$, denote the corresponding source patches as $X_A^r$ and $X_B^r$. Let $S^r$ denote the latent useful scene information in this region. Since $S^r$ is unobservable in unsupervised fusion, we quantify modality-specific information using conditional mutual information:

$$u_A^r = I(S^r; X_A^r \mid X_B^r), \qquad u_B^r = I(S^r; X_B^r \mid X_A^r). \tag{3}$$

Here, $u_A^r$ and $u_B^r$ measure the useful information uniquely provided by modalities $A$ and $B$, respectively. The ideal local fusion utility is defined as the joint preservation of both modality-specific components:

$$\mathcal{U}^{r,\star} = u_A^r + u_B^r. \tag{4}$$

Given a local contribution decision $Z^r$, let $\rho_A(Z^r) \in [0, 1]$ and $\rho_B(Z^r) \in [0, 1]$ denote the preservation ratios of useful information from modalities $A$ and $B$, respectively. The retained utility under decision $Z^r$ is

$$\mathcal{U}^r(Z^r) = \rho_A(Z^r)u_A^r + \rho_B(Z^r)u_B^r. \tag{5}$$

Therefore, the decision-induced information regret is

$$\begin{aligned} \mathcal{R}_{\text{dec}}^r(Z^r) &= \mathcal{U}^{r,\star} - \mathcal{U}^r(Z^r) \\ &= (1 - \rho_A(Z^r))u_A^r + (1 - \rho_B(Z^r))u_B^r. \end{aligned} \tag{6}$$

Let the modality-specific suppression ratios be

$$\varepsilon_A^r = 1 - \rho_A(Z^r), \qquad \varepsilon_B^r = 1 - \rho_B(Z^r). \tag{7}$$

Then Eq. (6) can be rewritten as

$$\mathcal{R}_{\text{dec}}^r(Z^r) = \varepsilon_A^r u_A^r + \varepsilon_B^r u_B^r. \tag{8}$$

To connect regret with total modality-specific information, we define the effective contribution error level:

$$\varepsilon_{\text{eff}}^r = \frac{\varepsilon_A^r u_A^r + \varepsilon_B^r u_B^r}{u_A^r + u_B^r}. \tag{9}$$

Thus,

$$\mathcal{R}_{\text{dec}}^r(Z^r) = \varepsilon_{\text{eff}}^r \mathcal{U}^{r,\star} = \varepsilon_{\text{eff}}^r(u_A^r + u_B^r), \tag{10}$$

and

$$\frac{\partial \mathcal{R}_{\text{dec}}^r}{\partial \varepsilon_{\text{eff}}^r} = \mathcal{U}^{r,\star} = u_A^r + u_B^r. \tag{11}$$

This indicates that, under the same effective contribution error level, regions with larger $u_A^r + u_B^r$ suffer larger information regret. Therefore, discrepancy or complementary regions are decision-sensitive: incorrect contribution allocation in these regions is more likely to suppress useful modality-specific information and cause substantial information loss. Detailed derivations are provided in Appendix D.1.

## 2.2. Observation-Conditioned Decision Learning

Existing fusion networks may include rich internal mechanisms, such as encoded features, attention maps, gating activations, semantic priors, or foundation-model representations. We do not assume that these representations are uninformative. Instead, the limitation is that such implicit states are not explicitly constrained to separate decision-relevant factors, such as cross-modal discrepancy, complementary evidence, reliability, and decision uncertainty. As shown in Appendix D.3, this may lead to non-negligible decision risk in discrepancy-sensitive regions when the policy remains over-confident under ambiguous observations.

To reduce such decision ambiguity, we construct explicit decision-relevant observations:

$$\Pi = \Psi(X) \in \mathcal{P}. \tag{12}$$

Here, $\Pi$ is not the raw source image itself, nor a generic hidden feature. Instead, it denotes explicit decision conditions extracted from multimodal inputs, including discrepancy, complementary evidence, and reliability. Given $\Pi$, we define an observation-conditioned decision distribution:

$$p_\alpha(Z \mid \Pi), \tag{13}$$

where $\alpha$ denotes the learnable parameters of the observation-to-decision mapping. This distribution encodes which modality contribution policies are plausible under the current discrepancy, complementarity, and reliability conditions. Therefore, the contribution decision is not an unconditional deterministic activation, but an explicit decision induced by observations.

## 2.3. Evidence-Calibrated Contribution Policy

For two-modality fusion, we instantiate $Z^r$ as a continuous local contribution weight. Let $Z_B^r \in [0, 1]$ denote the contribution of modality $B$, and $1 - Z_B^r$ denote the contribution of modality $A$. The observation $\Pi^r$ is used to construct modality-specific evidence $e_A^r$ and $e_B^r$. In our method, these evidence terms are obtained from spatial discrepancy, spectral energy, complementary spectra, and reliability:

$$e_A^r = D_{\text{spa}}^r R_A^r (E_A^r + \eta C_A^r),$$
$$e_B^r = D_{\text{spa}}^r R_B^r (E_B^r + \eta C_B^r), \tag{14}$$

where $D_{\text{spa}}^r$ denotes spatial discrepancy, $E_A^r, E_B^r$ denote spectral energy, $C_A^r, C_B^r$ denote complementary spectral evidence, and $R_A^r, R_B^r$ denote reliability estimates. Thus, $e_A^r$ and $e_B^r$ summarize how strongly each modality supports the local contribution decision under the current observation.

We parameterize the local contribution decision with a symmetric Beta prior:

$$Z_B^r \sim \text{Beta}(e_B^r + \lambda, e_A^r + \lambda), \tag{15}$$

where $\lambda > 0$ is a pseudo-count that introduces a conservative prior when observational evidence is weak. The mean of this distribution gives the contribution policy:

$$W_B^r = \mathbb{E}[Z_B^r \mid \Pi^r] = \frac{e_B^r + \lambda}{e_A^r + e_B^r + 2\lambda}. \tag{16}$$

The relative evidence determines the direction of modality allocation. Indeed,

$$W_B^r - \frac{1}{2} = \frac{e_B^r + \lambda}{e_A^r + e_B^r + 2\lambda} - \frac{1}{2}$$
$$= \frac{e_B^r - e_A^r}{2(e_A^r + e_B^r + 2\lambda)}. \tag{17}$$

Since the denominator is positive, we have

$$W_B^r > \frac{1}{2} \iff e_B^r > e_A^r, \qquad W_B^r < \frac{1}{2} \iff e_B^r < e_A^r. \tag{18}$$

Thus, when the evidence is reliable, the induced contribution policy is direction-consistent.

Decision confidence is controlled by the total evidence, and we quantify evidence insufficiency as:

$$U^r = \frac{2\lambda}{e_A^r + e_B^r + 2\lambda}. \tag{19}$$

The final calibrated contribution policy is

$$\bar{W}_B^r = (1 - U^r)W_B^r + \frac{1}{2}U^r. \tag{20}$$

This calibration shrinks uncertain decisions toward conservative mixing:

$$\bar{W}_B^r - \frac{1}{2} = (1 - U^r)\left(W_B^r - \frac{1}{2}\right). \tag{21}$$

Therefore, uncertainty modulation does not change the evidence-induced decision direction, but only attenuates its magnitude. When observations provide sufficient evidence, $U^r$ is small and $\bar{W}_B^r$ approaches $W_B^r$. When observations are insufficient or unreliable, $U^r$ becomes large and $\bar{W}_B^r$ approaches $0.5$, corresponding to conservative mixing. This property prevents unreliable observations from producing over-confident modality allocation in discrepancy regions. Additional analysis of the monotonicity of $U^r$ is provided in Appendix D.2.

The calibrated contribution policy is then applied to modulate modality-specific feature tensors:

$$\widehat{\mathbf{F}}_B^r = \bar{W}_B^r \odot \mathbf{F}_B^r,$$
$$\widehat{\mathbf{F}}_A^r = (1 - \bar{W}_B^r) \odot \mathbf{F}_A^r. \tag{22}$$

The fused image is reconstructed from these decision-conditioned features:

$$\mathbf{Y} = \mathcal{F}_\omega(\mathbf{X}, \mathbf{Z}), \qquad \mathbf{Z} = \{\bar{W}_B^r\}_r. \tag{23}$$

In Appendix D.3, we analyze the limitation of implicit deterministic policies.

## 3. Method

### 3.1. Overview

As shown in Figure 2, IDEAL is an observation-conditioned decision-learning framework consisting of an encoder, a Modal Difference Attention (MDA) module, a Conditional Decision Gating (CDG) module, and a decoder. The encoder extracts multi-scale modality features. MDA constructs spatial discrepancy observations, while CDG estimates evidence-calibrated contribution policies from spectral energy, complementary spectra, and spectral-entropy reliability. These policies modulate modality features before concatenation, and the decoder reconstructs the fused image from the resulting decision-conditioned features. The whole framework is optimized with unsupervised fusion losses for intensity and structural information preservation.

Specifically, we employ a convolutional encoder to extract four-scale features with channel numbers of 8, 16, 24, and 32. At each scale, feature extraction is performed by convolutions with kernel size 3, stride 2, and padding 1. To enrich the observation conditions for decision learning, we introduce two complementary modules. MDA computes cross-modal spatial feature differences and enhances discrepancy cues through attention, highlighting regions where local

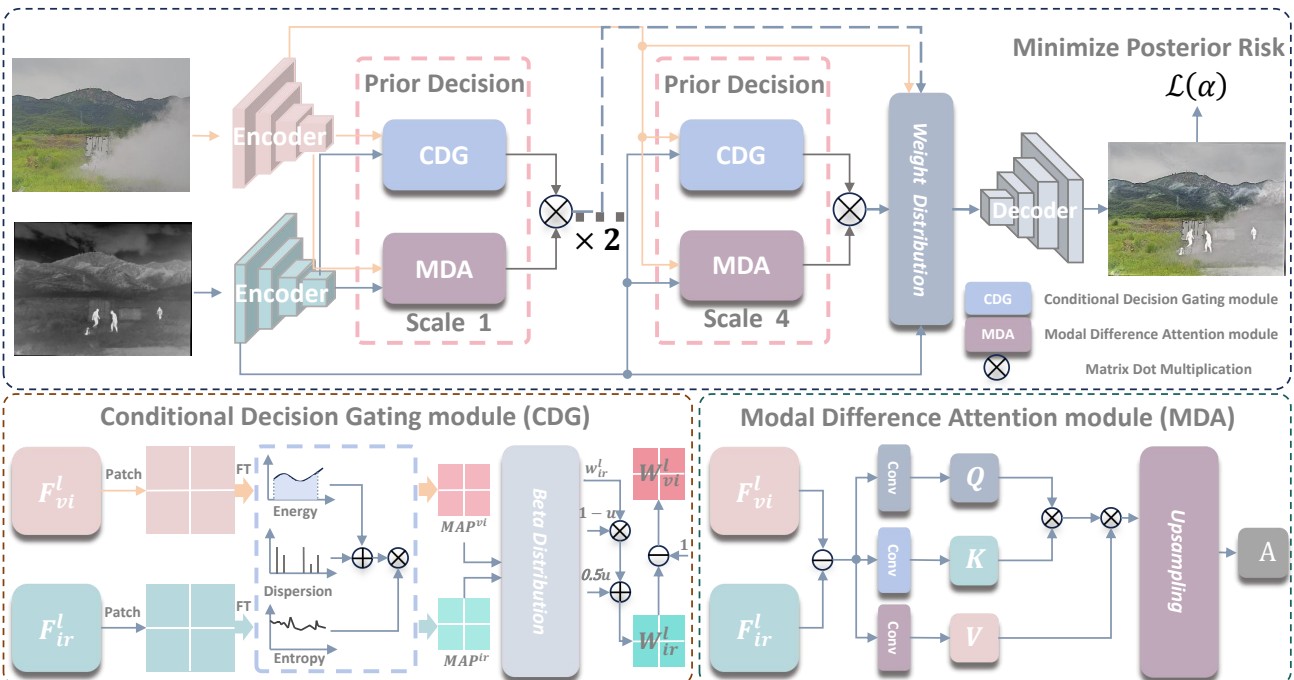

*Figure 2.* The framework of IDEAL. The network mainly consists of prior decision-making and posterior risk minimization. After the encoder extracts features, it generates a modal feature weight map through the MDA and CDG modules, which is then assigned to the modal features. Finally, the decoder upsamples the data to the fused image.

contribution decisions are needed while suppressing noisy variations. In parallel, CDG projects same-scale modality features into the frequency domain and derives decision evidence from power-spectrum energy, complementary spectra, and spectral-entropy reliability. The energy and complementary terms indicate which modality provides stronger local evidence, while the entropy-derived reliability determines whether the evidence should be trusted. Together, these observations guide calibrated modality contribution decisions in discrepancy-sensitive regions.

During decoding, the modulated cross-modal features are concatenated across scales. Starting from the deepest scale, the decoder progressively upsamples features by bicubic interpolation, concatenates them with features from the previous scale, and repeats this process until the fused image is reconstructed at the original resolution.

### 3.2. Conditional Decision Gating module

The CDG module serves a single core purpose, namely determining which modality should be preferred in a given region. Accordingly, transforming features from the spatial domain to the frequency domain makes it easier to extract discriminative cues required for this decision, thereby guiding the gating allocation more reliably.

For the input features $\mathbf{F} \in \{\mathbf{F}_A, \mathbf{F}_B\}$, we first partition them into patches and then apply a Fourier transform to map each patch to the frequency domain, from which we obtain the corresponding power spectrum $\mathbf{P}_A^l = \left|\mathcal{F}(\mathbf{F}_A^l)\right|^2 \in \mathbb{R}^{C \times P \times P}$, $\mathbf{P}_B^l = \left|\mathcal{F}(\mathbf{F}_B^l)\right|^2 \in \mathbb{R}^{C \times P \times P}$, respectively. Then, the power spectrum is averaged across channels and frequency points to obtain the power spectrum energy value. This process can be formalized as follows:

$$E_A^l = \frac{1}{CP^2} \sum_{c=1}^{C} \sum_{u,v} \mathbf{P}_A^l(u,v),$$
$$E_B^l = \frac{1}{CP^2} \sum_{c=1}^{C} \sum_{u,v} \mathbf{P}_B^l(u,v). \tag{24}$$

Where $(u,v)$ represents two-dimensional frequency domain coordinates and $P$ represents patch size. Then, the differences in the modal characteristic spectra are calculated to account for the complementarity of the frequency components.

$$C_A^l = \frac{1}{CP^2} \sum_{c,u,v} \max\Big(\mathbf{P}_A^l(u,v) - \mathbf{P}_B^l(u,v), 0\Big),$$
$$C_B^l = \frac{1}{CP^2} \sum_{c,u,v} \max\Big(\mathbf{P}_B^l(u,v) - \mathbf{P}_A^l(u,v), 0\Big). \tag{25}$$

Finally, we calculate the spectral entropy and use it as a basis for decision reliability. Firstly, the power spectrum of

each channel is normalized into a distribution.

$$\mathbf{p}_A^l(u,v) = \frac{\mathbf{P}_A^l(u,v)}{\sum_{u,v} \mathbf{P}_A^l(u,v)}, \mathbf{p}_B^l(u,v) = \frac{\mathbf{P}_B^l(u,v)}{\sum_{u,v} \mathbf{P}_B^l(u,v)}. \tag{26}$$

Then calculate the channel entropy.

$$\begin{aligned} H_A^l &= -\sum_{u,v} \mathbf{p}_A^l(u,v) \log\big(\mathbf{p}_A^l(u,v)\big), \\ H_B^l &= -\sum_{u,v} \mathbf{p}_B^l(u,v) \log\big(\mathbf{p}_B^l(u,v)\big). \end{aligned} \tag{27}$$

Next, take the average value of the channels $\hat{H}_A^l = \frac{1}{C}\sum_{c=1}^{C} H_A^l, \hat{H}_B^l = \frac{1}{C}\sum_{c=1}^{C} H_B^l$. Finally, get the reliability as follows:

$$r_A^l = \exp\left(-\gamma \hat{H}_A^l\right), \qquad r_B^l = \exp\left(-\gamma \hat{H}_B^l\right). \tag{28}$$

The energy, complementary terms, spectral-entropy reliability, and modal differential attention are fused into patch-level decision evidence. Let $n \in \{1, \ldots, N_l\}$ index the unfolded patches at scale $l$. For the modal differential attention map $\mathbf{A}_l$, we first compute its patch-level average:

$$d_n^l = \frac{1}{P^2} \sum_{(x,y) \in \Omega_n^l} \mathbf{A}_l(x,y), \tag{29}$$

where $\Omega_n^l$ denotes the spatial support of the $n$-th patch. Then the patch-level evidence is computed as

$$\begin{aligned} e_{A,n}^l &= d_n^l \, r_{A,n}^l \left(E_{A,n}^l + \eta \, C_{A,n}^l\right), \\ e_{B,n}^l &= d_n^l \, r_{B,n}^l \left(E_{B,n}^l + \eta \, C_{B,n}^l\right). \end{aligned} \tag{30}$$

Here, $E_{m,n}^l$, $C_{m,n}^l$, and $r_{m,n}^l$ are patch-level scalar statistics for modality $m \in \{A, B\}$, while $d_n^l$ is the patch-level average of the modal differential attention map. The coefficient $\eta$ is a hyperparameter controlling the strength of complementary spectral evidence.

The patch-level weighting coefficient and uncertainty coefficient are obtained through beta-symmetric prior smoothing:

$$\begin{aligned} w_{B,n}^l &= \frac{e_{B,n}^l + \lambda}{e_{A,n}^l + e_{B,n}^l + 2\lambda}, \\ u_n^l &= \frac{2\lambda}{e_{A,n}^l + e_{B,n}^l + 2\lambda}. \end{aligned} \tag{31}$$

Following the implementation, the patch-level weight is first calibrated by uncertainty:

$$\widetilde{w}_{B,n}^l = (1 - u_n^l) w_{B,n}^l + \frac{1}{2} u_n^l. \tag{32}$$

After obtaining $\widetilde{w}_{B,n}^l$ and $u_n^l$ for all patches, they are backfilled into the feature pixel map through a fold-and-average operation:

$$\widetilde{\mathbf{W}}_B^l = \mathcal{B}\left(\{\widetilde{w}_{B,n}^l\}_{n=1}^{N_l}\right), \qquad \mathbf{U}^l = \mathcal{B}\left(\{u_n^l\}_{n=1}^{N_l}\right), \tag{33}$$

$\mathcal{B}(\cdot)$ denotes the fold-and-average backfilling operator.

The backfilled weight map is further calibrated by the uncertainty map:

$$\mathbf{W}_B^l = (\mathbf{1} - \mathbf{U}^l) \odot \widetilde{\mathbf{W}}_B^l + \frac{1}{2}\mathbf{U}^l. \tag{34}$$

The calibrated contribution policy is then applied to the modality features:

$$\begin{aligned} \hat{\mathbf{F}}_B^l &= \mathbf{F}_B^l \odot \mathbf{W}_B^l, \\ \hat{\mathbf{F}}_A^l &= \mathbf{F}_A^l \odot (\mathbf{1} - \mathbf{W}_B^l). \end{aligned} \tag{35}$$

When the patch-level evidence is insufficient, the uncertainty term pulls the contribution policy toward $0.5$, preventing unreliable patch conditions from producing over-confident modality allocation.

### 3.3. Modal Difference Attention module

The CDG module ensures that the system can make correct decisions in a certain region. However, CDG can only answer the question of *which region is favored more*, but cannot solve the problem of which regions should be given more importance in the entire feature map. Therefore, we introduced the MDA module.

In the MDA module, the feature map spectrum is first subjected to differential calculation. Simply performing a dot product between the difference graph and the decision weight graph will inevitably introduce a lot of noise. Therefore, we use three different projection convolutions to obtain the $\mathbf{Q}$, $\mathbf{K}$, and $\mathbf{V}$ tensors. These three tensors are then calculated as follows:

$$\mathbf{A}_l = \mathrm{softmax}\left(\frac{\mathbf{Q}\mathbf{K}^\top}{\sqrt{d}}\right) \cdot \mathbf{V}, \tag{36}$$

where $d$ is the scaling factor. This operation can obtain an enhanced contextual difference representation and filter out noise information. Finally, upsampling with a 1x1 convolution kernel restores the difference map to the same size as the feature map.

### 3.4. Loss Function

Since ground-truth fusion images are often unavailable for fusion tasks, the training objective focuses more on preserving cross-modal complementary information and suppressing conflicts, rather than fitting the semantic appearance of a single domain. To avoid introducing domain bias from pre-trained perceptual networks, we employ lightweight and

*Table 1.* Quantitative comparison on the MSRS dataset. The best result in each column is in **bold** and the second best is underlined.

| Method | EN↑ | SF↑ | AG↑ | SD↑ | SCD↑ | Params/M↓ | GFLOPs↓ | Memory/G↓ |
|---|---|---|---|---|---|---|---|---|
| YDTR (Tang et al., 2022b) | 5.7340 | 7.9270 | 2.3604 | 28.0495 | 1.1608 | 0.106 | 38.91 | 3.71 |
| CDDFuse (Zhao et al., 2023b) | 6.8004 | 12.2786 | 3.9663 | 48.0064 | **1.6665** | 1.19 | 220.92 | 16.93 |
| LRRNet (Li et al., 2023) | 6.3354 | 9.2960 | 2.9019 | 35.8350 | 0.8598 | **0.048** | 5.71 | 8.12 |
| CrossFuse (Li & Wu, 2024) | 6.6362 | 10.4545 | 3.3010 | 40.6893 | 1.0852 | 12.00 | 51.88 | 1.11 |
| EMMA (Zhao et al., 2024a) | 6.8215 | 12.4250 | 4.0729 | 49.4932 | 1.6516 | 1.52 | 16.75 | 1.17 |
| SAGE (Wu et al., 2025) | 6.1374 | 11.1985 | 3.4366 | 40.0595 | 1.4749 | 0.136 | 8.20 | 0.55 |
| TextFusion (Cheng et al., 2025b) | 6.1701 | 10.7431 | 3.1538 | 42.1810 | 1.4870 | 0.206 | 24.63 | 0.99 |
| GIFNet (Cheng et al., 2025a) | 6.0105 | **13.4107** | 3.5993 | 35.5469 | 1.4126 | 0.725 | 91.21 | 4.74 |
| IDEAL | **7.0327** | 12.6222 | **4.4290** | **50.9291** | 1.6649 | 0.18 | **1.00** | **0.37** |

*Table 2.* Quantitative comparison on the M3FD and FMB datasets. The best and second results are marked in **bold** and underlined.

| Method | M3FD | | | | | FMB | | | | |
|---|---|---|---|---|---|---|---|---|---|---|
| | EN↑ | SF↑ | AG↑ | SD↑ | SCD↑ | EN↑ | SF↑ | AG↑ | SD↑ | SCD↑ |
| YDTR | 6.4335 | 8.0243 | 2.4291 | 25.2569 | 1.4525 | 6.4341 | 9.6710 | 2.9330 | 27.5427 | 1.4575 |
| CDDFuse | 6.7465 | 11.7874 | 3.5914 | 32.8690 | 1.5853 | 6.7868 | 14.0264 | 4.2144 | 37.3003 | 1.6615 |
| LRRNet | 6.3313 | 8.4399 | 2.6287 | 23.9724 | 1.3813 | 6.3167 | 10.0430 | 3.1035 | 26.7392 | 1.3544 |
| CrossFuse | 6.3223 | 9.3751 | 2.7890 | 24.0031 | 0.8134 | 6.5015 | 12.3736 | 3.7539 | 31.0277 | 0.9561 |
| EMMA | 6.7496 | 12.3251 | 3.9788 | 33.0277 | 1.3995 | 6.7854 | 14.4790 | 4.5972 | 37.3016 | 1.5318 |
| SAGE | 6.7368 | 10.4625 | 3.2183 | 33.1467 | 1.6600 | 6.8144 | 12.1252 | 3.7185 | 36.8788 | 1.7301 |
| TextFusion | 6.5470 | 11.5236 | 3.4867 | 27.8155 | 1.5982 | 6.6351 | 13.9946 | 4.2031 | 31.8542 | 1.6376 |
| GIFNet | 6.8959 | **14.6138** | 4.1588 | 35.4071 | **1.7170** | 6.8761 | **17.8743** | **4.9269** | 37.6700 | **1.7492** |
| IDEAL | **7.1114** | 14.0942 | **4.6504** | **40.3091** | 1.5238 | **7.0147** | 14.5579 | 4.7895 | **41.7446** | 1.6312 |

task-independent reconstruction constraints as risk function to highlight the superiority of our prior decisions.

In summary, our risk function is as follows:

$$\mathcal{L}_{\text{fusion}} = \kappa \cdot \mathcal{L}_{int} + \varrho \cdot \mathcal{L}_{\text{grad}}. \tag{37}$$

Where $L_{int}$ is the pixel fidelity risk function and $L_{gra}$ is the gradient fidelity risk function. Their specific definitions as:

$$\mathcal{L}_{int} = \|\mathbf{I}_F - \max(\mathbf{I}_A, \mathbf{I}_B)\|_2. \tag{38}$$

$$\mathcal{L}_{\text{grad}} = \|\nabla_x \mathbf{I}_F - \max(\nabla_x \mathbf{I}_A, \nabla_x \mathbf{I}_B)\|_1 \\ + \|\nabla_y \mathbf{I}_F - \max(\nabla_y \mathbf{I}_A, \nabla_y \mathbf{I}_B)\|_1. \tag{39}$$

where $\nabla$ denotes the Sobel gradient operator, and $\max(\cdot)$ denotes selecting the maximum value within parentheses.

## 4. Experiments

For details regarding our experimental setup, please see the Appendix B. To evaluate fusion performance, we adopt five quantitative metrics: Entropy (EN), Spatial Frequency (SF), Average Gradient (AG), Standard Deviation (SD), and the Sum of Correlations of Differences (SCD). EN measures the amount of information contained in the fused image. SF reflects overall activity and texture richness. AG characterizes edge sharpness through the average gradient magnitude.

SD indicates global contrast and intensity dispersion. SCD assesses the extent to which complementary information in the source images is preserved in the fusion result. In general, higher values of these metrics correspond to better fusion quality.

### 4.1. Comparison Experiment

We present detailed results comparing our method with existing approaches on the MSRS (Tang et al., 2022a), M3FD (Liu et al., 2022), and FMB (Liu et al., 2023) datasets to provide a comprehensive quantitative evaluation, as summarized in Table 1 and Table 2.

We selected the most advanced methods from 2023 to 2025 for quantitative and visualization analysis. These methods are YDTR (Tang et al., 2022b), CDDFuse (Zhao et al., 2023b), LRRNet (Li et al., 2023), CrossFuse (Li & Wu, 2024), EMMA (Zhao et al., 2024a), SAGE (Wu et al., 2025), TextFusion (Cheng et al., 2025b), and GIFNet (Cheng et al., 2025a). We cropped the MSRS dataset to 352×352 for computational complexity analysis.

As shown in Figure 4, IDEAL uses discrepancy and reliability cues to produce spatially adaptive modality contribution decisions. This enables it to suppress halos, artifacts, and modality bias while preserving visible textures and enhanc-

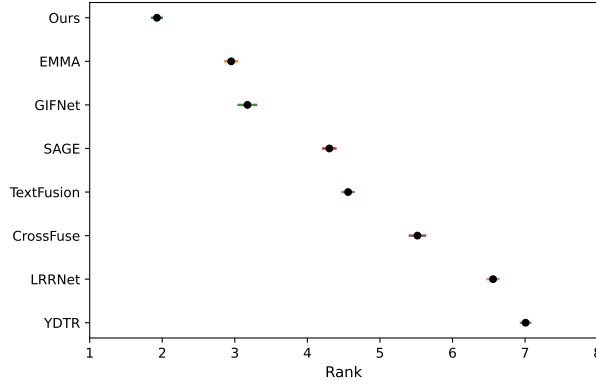

*(a) Average-rank visualization.*

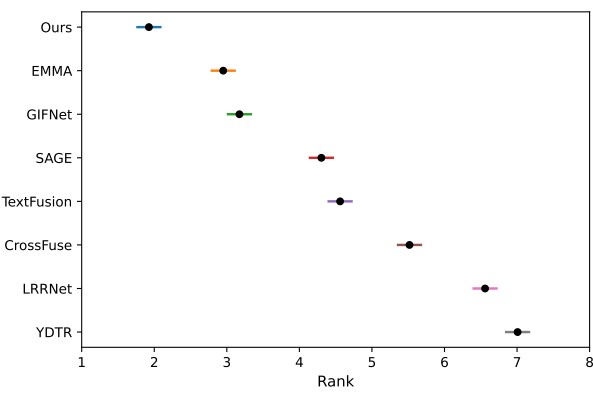

*(b) Nemenyi post-hoc visualization.*

Figure 3. Friedman–Nemenyi statistical visualization over image-metric blocks. Black dots denote average ranks, and horizontal bars indicate critical-distance-based intervals. A smaller rank indicates better overall performance.

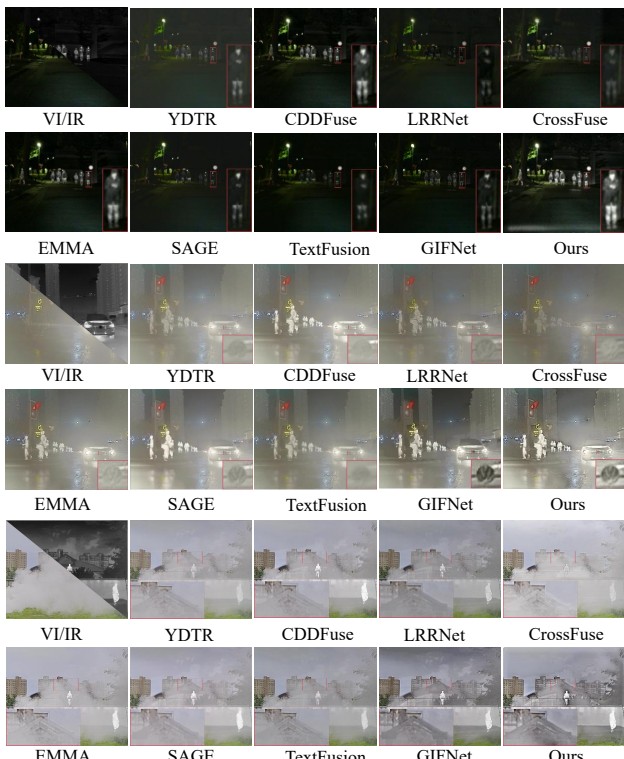

Figure 4. Comparison results on different datasets. From top to bottom, the datasets are MSRS, M3FD, and FMB.

ing thermal targets across nighttime, hazy, and daytime scenes. In contrast, the baseline method is more sensitive to cross-modal conflicts and often exhibits excessive dulling and loss of detail and texture.

As shown in Table 1 and Table 2, our method achieves consistently strong and balanced fusion performance on MSRS, M3FD, and FMB. IDEAL obtains the best EN and SD on all three datasets, and achieves the best AG on MSRS and M3FD and the second-best AG on FMB. These results indicate that the fused images produced by IDEAL contain richer information, stronger contrast, and clearer structural details under diverse imaging conditions. Although some baselines obtain higher SF or SCD by amplifying high-frequency responses, such gains are often accompanied by over-sharpening, artifacts, or modality bias. In contrast, IDEAL uses discrepancy-aware observations and evidence-calibrated contribution policies to preserve informative structures while suppressing unreliable responses, leading to better EN, AG, and SD with competitive SCD. Notably, these improvements are achieved without increas-

ing model complexity: on MSRS, IDEAL uses only 0.18M parameters, 1.00 GFLOPs, and 0.37G memory, while still delivering the strongest overall quantitative performance. This highlights the effectiveness of explicit contribution decision learning without a heavier fusion backbone.

More experimental results on multimodal image fusion can be found in the Appendix C.

### 4.2. Friedman–Nemenyi Statistical Analysis

To further assess the statistical consistency of the quantitative results in Tables 1 and 2, we perform the Friedman test (Demšar, 2006) followed by the Nemenyi post-hoc test (Nemenyi, 1963). Each image-metric pair is used as one evaluation block, covering EN, SF, AG, SD, and SCD on MSRS, M3FD, and FMB, resulting in 910 blocks. The Friedman test rejects the null hypothesis of equivalent average ranks with $\chi^2_F = 3395.96$ and $p < 0.001$, and the Iman-Davenport correction further confirms significant differences with $F_F = 1037.96$ and $p < 0.001$. As shown in Fig. 3, our method obtains the lowest average rank of 1.93, and the Nemenyi post-hoc visualization with a critical distance of 0.35 at $\alpha = 0.05$ shows a clear separation from the baselines. This demonstrates that our improvements are statistically consistent with the results.

*Table 3.* Ablation study on MSRS, M3FD, and FMB. The best result in each column is in **bold** and the second best is underlined.

| Method | MSRS | | | | | M3FD | | | | | FMB | | | | |
|---|---|---|---|---|---|---|---|---|---|---|---|---|---|---|---|
| | EN↑ | SF↑ | AG↑ | SD↑ | SCD↑ | EN↑ | SF↑ | AG↑ | SD↑ | SCD↑ | EN↑ | SF↑ | AG↑ | SD↑ | SCD↑ |
| *Model I* | 6.73 | 11.61 | 3.88 | 46.00 | 1.66 | 6.69 | 11.16 | 3.68 | 30.98 | 1.52 | 6.73 | 12.55 | 4.10 | 35.00 | 1.61 |
| *Model II* | 6.73 | 10.81 | 3.46 | 46.47 | 1.60 | 6.80 | 11.10 | 3.55 | 32.90 | 1.05 | 6.83 | 11.84 | 3.88 | 36.29 | 1.29 |
| *Model III* | 6.86 | 12.10 | 4.12 | 46.07 | 1.66 | 6.93 | 12.54 | 4.20 | 35.93 | 1.51 | 7.00 | 13.73 | 4.55 | 38.14 | 1.62 |
| *Model IV* | 6.76 | 11.96 | 4.06 | 44.33 | 1.61 | 6.91 | 13.45 | 4.64 | 34.56 | 1.33 | 6.93 | 14.55 | **5.03** | 37.80 | 1.47 |
| *Full Model* | **7.03** | **12.62** | **4.43** | **50.93** | 1.66 | **7.11** | **14.09** | **4.65** | **40.31** | 1.52 | **7.01** | **14.56** | 4.79 | **41.74** | **1.63** |

*Table 4.* Detection performance comparison of different methods on MSRS dataset. The best and the second results are marked in **bold** and underlined.

| Method | Precision | Recall | mAP@0.5 | mAP@[0.5:0.95] |
|---|---|---|---|---|
| VIS | 91.3 | 76.5 | 80.7 | 53.3 |
| IR | 88.2 | 69.1 | 81.6 | 57.0 |
| YDTR | 90.4 | **88.7** | 91.9 | 65.7 |
| CDDFuse | 91.2 | 86.6 | 91.7 | 64.7 |
| LRRNet | **94.8** | 85.2 | 91.5 | 66.9 |
| CrossFuse | 91.5 | 82.0 | 89.4 | 60.4 |
| EMMA | 91.1 | 84.2 | 90.9 | 64.3 |
| SAGE | 94.7 | 86.3 | **93.3** | 65.0 |
| TextFusion | 90.7 | 79.7 | 87.6 | 57.1 |
| GIFNet | 93.4 | 85.5 | 92.6 | 65.5 |
| Ours | 93.7 | 85.0 | 93.0 | **67.5** |

### 4.3. Ablation

To evaluate the importance of each component in the prior decision-making and posterior risk minimization phases, we conducted targeted ablation experiments, which were divided into *Model I,II, III, IV*. *Model I*: eliminate MDA and CDG module. *Model II*: eliminate gradient risk function. *Model III*: eliminate pixel intensity risk function. *Model IV*: eliminate MDA module.

The ablation results consistently validate the complementary roles of different components. The intensity-loss variant mainly improves global information preservation, while the gradient-loss variant enhances structural sharpness. The MDA and CDG modules further contribute to region-wise discrepancy perception and evidence-calibrated modality allocation. The full IDEAL model achieves the most balanced performance across MSRS, M3FD, and FMB, validating observation-conditioned contribution decisions in reducing modality bias and artifacts.

### 4.4. Downstream Task

As reported in Table 4, our method achieves the best performance on the most stringent metric, mAP@[0.5:0.95] (**67.5**), indicating stronger localization quality and better robustness across IoU thresholds. Although several competitors obtain the best result on individual metrics, *e.g.,*, LRRNet achieves the highest Precision (**94.8**), YDTR obtains the

highest Recall (**88.7**), and SAGE leads on mAP@0.5 (**93.3**), our method remains highly competitive in Precision/Recall (93.7/85.0) and achieves the second-best mAP@0.5 (93.0). These results show that IDEAL's observation-conditioned decisions improve high-IoU detection through stable, conflict-aware modality allocation.

## 5. Conclusion

This paper introduces IDEAL, an observation-conditioned decision-learning framework for multimodal image fusion. IDEAL reformulates fusion as explicit local contribution decision-making, where cross-modal discrepancy, complementary evidence, and reliability cues are transformed into decision-relevant observations. Through modal difference attention and evidence-calibrated contribution gating, IDEAL enables adaptive and uncertainty-aware modality allocation, reducing modality bias and artifact propagation in discrepancy-sensitive regions. Extensive experiments across diverse fusion tasks and datasets demonstrate its strong and robust performance against state-of-the-art methods. Nevertheless, current decision observations mainly rely on spatial discrepancy and frequency-domain statistics. Future work will explore semantic cues, task feedback, and temporal consistency for broader multimodal fusion scenarios.

## Acknowledgement

This work was supported by the National Natural Science Foundation of China (Grant no. 62576298), the Sichuan Science and Technology Program (Grant no. 2025ZNS-FSC0474), the Mianyang Science and Technology Program (Grant no. 2025ZYDF096), and the Outstanding Talent Cultivation and Introduction Program (Grant no. 25ZX7182), Postgraduate Innovation Fund Project by Southwest University of Science and Technology (Grant no. 26ycx2057).

## Impact Statement

This paper presents work whose goal is to advance the field of Machine Learning. There are many potential societal consequences of our work, none of which we feel must be specifically highlighted here.

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

The following appendices provide a glossary of symbols, further details regarding the experimental setup and results, and a more detailed account of our derivation process. Specifically, these appendices are organized as follows: first, a glossary of symbols (Appendix A); followed by details on the experimental setup (Appendix B) and results obtained on medical images, multi-exposure images, and multi-focus images (Appendix C); and finally, additional details regarding the derivation process (Appendix D).

## A. Notations

*Table 5.* Summary of major notations used in this paper.

| Notation | Type | Description |
| --- | --- | --- |
| **General formulation** | | |
| $X = \{X^1, \ldots, X^M\}$ | Tensor set | Multimodal input containing $M$ source modalities. |
| $X_A, X_B$ | Tensor | Two source modalities in the two-modality formulation. |
| $Y$ | Tensor | Fused output image. |
| $\mathcal{Y}$ | Set | Output space of the fused image. |
| $Z$ | Decision variable | Modality contribution decision, representing internal fusion controls such as gating, attention, modulation, or modality selection. |
| $\mathcal{Z}$ | Set | Decision space of $Z$. |
| $\mathcal{F}_\omega(X, Z)$ | Function | Decision-conditioned fusion mapping parameterized by $\omega$. |
| $\omega$ | Parameter | Learnable parameters of the fusion operator. |
| $\Pi = \Psi(X)$ | Observation | Explicit decision-relevant observation extracted from multimodal inputs. |
| $\mathcal{P}$ | Set | Observation space of $\Pi$. |
| $p_\alpha(Z \mid \Pi)$ | Distribution | Observation-conditioned decision distribution. |
| $\alpha$ | Parameter | Learnable parameters of the observation-to-decision mapping. |
| **Decision sensitivity analysis** | | |
| $r$ | Index | Local region or local patch index. |
| $S^r$ | Latent variable | Latent useful scene information in region $r$. |
| $u_A^r, u_B^r$ | Scalar | Modality-specific useful information uniquely provided by modalities $A$ and $B$. |
| $\mathcal{U}^{r,\star}$ | Scalar | Ideal local fusion utility, defined as $u_A^r + u_B^r$. |
| $\mathcal{U}^r(Z^r)$ | Scalar | Retained local utility under decision $\mathbf{Z}^r$. |
| $\rho_A(Z^r), \rho_B(Z^r)$ | Scalar | Preservation ratios of useful information from modalities $A$ and $B$. |
| $\mathcal{R}_{\text{dec}}^r(Z^r)$ | Scalar | Decision-induced information regret in region $r$. |
| $\varepsilon_A^r, \varepsilon_B^r$ | Scalar | Modality-specific information suppression ratios. |
| $\varepsilon_{\text{eff}}^r$ | Scalar | Effective contribution error level. |
| **Conditional Decision Gating module** | | |
| $l$ | Index | Network scale index. |
| $n$ | Index | Patch index after unfolding the feature map. |
| $N_l$ | Integer | Number of unfolded patches at scale $l$. |
| $\mathbf{F}_A^l, \mathbf{F}_B^l$ | Tensor | Feature maps of modalities $A$ and $B$ at scale $l$. |
| $\mathbf{X}_{m,n}^l$ | Tensor | The $n$-th local patch of modality $m \in \{A, B\}$ at scale $l$. |
| $C_l$ | Integer | Number of feature channels at scale $l$. |
| $P$ | Integer | Patch size used in the FFT-based evidential gate. |
| $(u, v)$ | Index | Two-dimensional frequency-domain coordinates. |
| $\mathcal{F}_{\text{2D}}(\cdot)$ | Operator | Two-dimensional Fourier transform. |
| $\mathbf{P}_{m,n}^l$ | Tensor | Patch-wise power spectrum of modality $m$. |
| $E_{m,n}^l$ | Scalar | Patch-level power-spectrum energy of modality $m$. |

| Notation | Type | Description |
| --- | --- | --- |
| $C_{m,n}^l$ | Scalar | Patch-level complementary spectral evidence of modality $m$. |
| $\mathbf{p}_{m,n}^l$ | Tensor | Normalized power-spectrum distribution of modality $m$. |
| $H_{m,n}^l$ | Scalar | Patch-level spectral entropy of modality $m$. |
| $r_{m,n}^l$ | Scalar | Patch-level reliability score computed from spectral entropy. |
| $\gamma$ | Hyperparameter | Sensitivity coefficient for entropy-based reliability estimation. |
| $\mathbf{A}^l$ | Tensor | Modal differential attention map at scale $l$. |
| $\Omega_n^l$ | Set | Spatial support of the $n$-th patch at scale $l$. |
| $d_n^l$ | Scalar | Patch-level average of the modal differential attention map. |
| $e_{A,n}^l, e_{B,n}^l$ | Scalar | Patch-level decision evidence for modalities $A$ and $B$. |
| $\eta$ | Hyperparameter | Weight controlling the strength of complementary spectral evidence. |
| $\lambda$ | Hyperparameter | Pseudo-count of the symmetric Beta prior. |
| $w_{B,n}^l$ | Scalar | Patch-level contribution weight of modality $B$ before uncertainty calibration. |
| $u_n^l$ | Scalar | Patch-level uncertainty coefficient. |
| $\widetilde{w}_{B,n}^l$ | Scalar | Patch-level contribution weight after uncertainty calibration. |

**Backfilling, feature modulation, and modal difference attention**

| | | |
| --- | --- | --- |
| $\mathcal{B}(\cdot)$ | Operator | Fold-and-average backfilling operator that maps patch-level values to a spatial map. |
| $\widetilde{\mathbf{W}}_B^l$ | Tensor | Backfilled contribution weight map before the final uncertainty calibration. |
| $\mathbf{U}^l$ | Tensor | Backfilled uncertainty map at scale $l$. |
| $\mathbf{W}_B^l$ | Tensor | Final calibrated contribution weight map for modality $B$. |
| $\mathbf{1}$ | Tensor | All-one tensor with the same spatial size as the corresponding weight or uncertainty map. |
| $\widehat{\mathbf{F}}_A^l, \widehat{\mathbf{F}}_B^l$ | Tensor | Modality features after contribution weighting. |
| $\epsilon$ | Scalar | Small numerical constant for stable division and logarithm computation. |
| $\mathbf{D}^l$ | Tensor | Cross-modal spatial feature difference at scale $l$. |
| $\mathbf{Q}^l, \mathbf{K}^l, \mathbf{V}^l$ | Tensor | Query, key, and value tensors used in the modal difference attention module. |
| $d_h$ | Scalar | Head dimension used as the attention scaling factor. |
| $\mathbf{C}_{\text{ctx}}^l$ | Tensor | Contextual difference representation produced by attention. |
| $\sigma(\cdot)$ | Function | Sigmoid function used to generate the final attention map. |
| $\phi_q, \phi_k, \phi_v, \phi_o, \phi_d, \phi_m$ | Function | Learnable projection functions, implemented by $1 \times 1$ convolutions. |

**Loss functions and evaluation metrics**

| | | |
| --- | --- | --- |
| $\mathcal{L}_{\text{fusion}}$ | Scalar | Overall fusion loss. |
| $\mathcal{L}_{\text{int}}$ | Scalar | Pixel-intensity fidelity loss. |
| $\mathcal{L}_{\text{gra}}$ | Scalar | Gradient fidelity loss. |
| $\kappa, \varrho$ | Hyperparameter | Weights of the intensity loss and gradient loss. |
| $\mathbf{I}_F$ | Tensor | Fused image used in the loss function. |
| $\mathbf{I}_A, \mathbf{I}_B$ | Tensor | Source images used in the loss function. |
| $\nabla_x, \nabla_y$ | Operator | Sobel gradient operators in horizontal and vertical directions. |
| EN | Metric | Entropy. |
| SF | Metric | Spatial frequency. |
| AG | Metric | Average gradient. |
| SD | Metric | Standard deviation. |
| SCD | Metric | Sum of correlations of differences. |

| Notation | Type | Description |
|---|---|---|
| PSNR | Metric | Peak signal-to-noise ratio. |
| MSE | Metric | Mean squared error. |
| Nabf | Metric | Artifact metric. |
| VIF | Metric | Visual information fidelity. |

## B. Experiment Details

During training, we used the MSRS (Tang et al., 2022a) dataset. It contains a diverse set of scene image pairs captured under both daytime and nighttime conditions. We used 1,083 image pairs for training and reserved 80 pairs for testing, all with a resolution of $640 \times 480$.

In Eq. (31) and Eq. (37), $\lambda$ is set to 1, $\kappa$ and $\varrho$ are both set to 10. The image fusion model is trained for 200 epochs to ensure sufficient optimization. The initial learning rate is set to $2 \times 10^{-3}$, and the weight decay is set to $1 \times 10^{-4}$. We use a batch size of 16 and the Adam optimizer. All training and testing are conducted on an NVIDIA GeForce RTX 3090 24GB.

In this experiment, we demonstrate the superiority of our method and validate its fusion performance through comparative experiments with eight state-of-the-art methods, all of which utilize the original pre-trained weights released by their respective authors. The comparisons are performed on the MSRS, M3FD (Liu et al., 2022), and FMB (Liu et al., 2023) datasets, each with sizes of 640×480, 1024×768, and $800 \times 600$, respectively. Theses three datasets includes two typical scenarios: daytime and nighttime. The M3FD and FMB have a large number of smoke scenes in both lighting conditions.

*Table 6.* Details of selected IVIF datasets used in our experiments.

| Dataset | Img pairs | Resolution | Color | Camera angle | Nighttime | Objects/Categories | Challenge Scenes | Annotation |
|---|---|---|---|---|---|---|---|---|
| MSRS | 1569 | 640×480 | ✓ | driving | 749 | abundant / 8 | ✗ | ✓ |
| M3FD | 4200 | 1024×768 | ✓ | multiplication | 1671 | 33603 / 6 | ✓ | ✓ |
| FMB | 1500 | 800×600 | ✓ | multiplication | 826 | abundant / 14 | ✗ | ✓ |

## C. Additional Multimodal Image Fusion Analysis

In light of the task characteristics of multimodal medical image fusion, we evaluate not only the information content of the fused results via entropy, but also conduct quantitative analysis using fidelity-oriented metrics such as PSNR, MSE, Nabf, and VIF. The experimental results show that our method maintains high entropy while achieving a higher peak signal-to-noise ratio and a lower mean squared error; meanwhile, perceptual quality indicators remain competitive with other approaches. These findings suggest that our difference-aware decision system enables strong performance across a broader range of multimodal fusion tasks. To further evaluate whether such performance gains are statistically consistent, we perform a Friedman test followed by the Nemenyi post-hoc test using image-level metric results over EN, PSNR, MSE, Nabf, and VIF. As shown in Fig. 7, the black dots denote the average ranks of different methods, where a smaller rank indicates better overall performance, and the horizontal bars represent critical-distance-based rank intervals. Our method obtains the lowest average rank, demonstrating the strongest overall ranking among the compared methods. Moreover, its clear separation from most baselines under the Nemenyi criterion suggests that the advantage of IDEAL is not limited to a single metric, but is consistently reflected across image-level evaluations on the medical fusion task.

Figure 5 and Figure 6 present qualitative comparisons on Multi-Exposure Fusion(MEF) (Zhang, 2021) and Multi-Focus Fusion (MFF). In these challenging cross-task settings, several competing methods exhibit noticeable modality bias, leading to incomplete preservation of salient details. In contrast, our decision-driven strategy maintains a more balanced integration of complementary information, yielding visually more faithful and stable fused results.

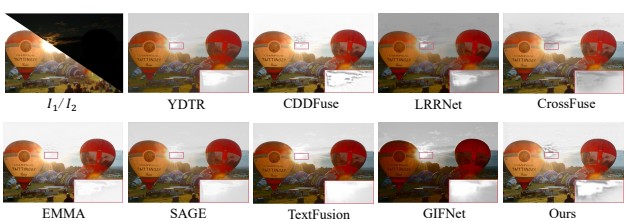

*Figure 5.* Visualization of fusion result on Multi-Exposure image Fusion (MEF) tasks.

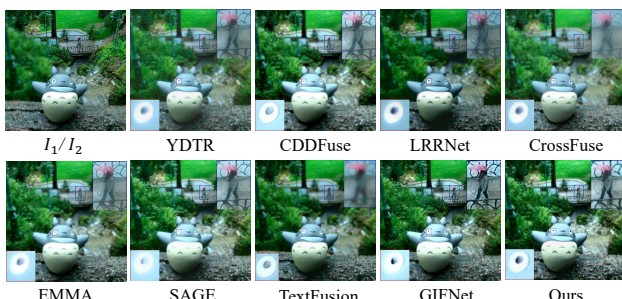

*Figure 6.* Visualization of fusion result on Multi-Focus image Fusion (MFF) tasks.

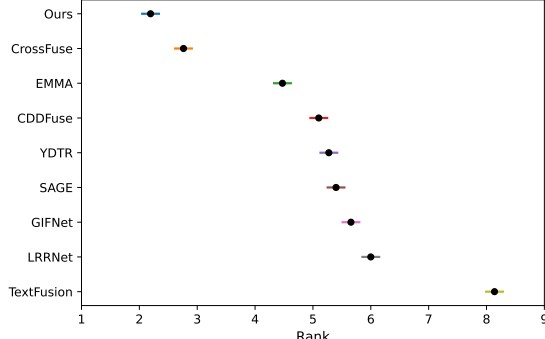

*Figure 7.* Nemenyi post-hoc visualization on the Harvard medical fusion dataset.

*Table 7.* Quantitative comparison on the Harvard dataset. The best result in each column is in **bold** and the second best is underlined.

| Method | EN↑ | PSNR↑ | MSE↓ | Nabf↓ | VIF↑ |
|---|---|---|---|---|---|
| YDTR | 5.7221 | 60.64 | 0.05663 | **0.00285** | 0.5032 |
| CDDFuse | 5.4561 | 61.32 | 0.04919 | 0.01222 | 0.6020 |
| LRRNet | 5.6505 | 61.15 | 0.05053 | 0.01300 | 0.4418 |
| CrossFuse | 5.7915 | 62.17 | 0.04009 | 0.00593 | **0.6128** |
| EMMA | 6.1080 | 61.04 | 0.05198 | 0.00703 | 0.5626 |
| SAGE | 5.6686 | 61.11 | 0.05130 | 0.00623 | 0.4810 |
| TextFusion | 5.5895 | 58.35 | 0.09604 | 0.01429 | 0.3831 |
| GIFNet | 5.4826 | **62.42** | **0.03777** | 0.03686 | 0.4145 |
| IDEAL | **6.8177** | 62.38 | 0.03800 | 0.00450 | 0.5329 |

# D. Additional Derivations for Decision Learning

### D.1. Derivation of Decision-Induced Information Regret

For a local region $r$, the ideal local fusion utility corresponds to jointly preserving the useful modality-specific information from both modalities:

$$\mathcal{U}^{r,\star} = u_A^r + u_B^r. \tag{40}$$

Under contribution decision $Z^r$, the retained useful information from modalities $A$ and $B$ is weighted by $\rho_A(Z^r)$ and $\rho_B(Z^r)$, respectively:

$$\mathcal{U}^r(Z^r) = \rho_A(Z^r)u_A^r + \rho_B(Z^r)u_B^r. \tag{41}$$

The decision-induced information regret is defined as

$$\mathcal{R}_{\text{dec}}^r(Z^r) = \mathcal{U}^{r,\star} - \mathcal{U}^r(Z^r). \tag{42}$$

Substituting the above definitions gives

$$\begin{aligned} \mathcal{R}_{\text{dec}}^r(Z^r) &= u_A^r + u_B^r - \rho_A(Z^r)u_A^r - \rho_B(Z^r)u_B^r \\ &= (1 - \rho_A(Z^r))u_A^r + (1 - \rho_B(Z^r))u_B^r. \end{aligned} \tag{43}$$

Let

$$\varepsilon_A^r = 1 - \rho_A(Z^r), \qquad \varepsilon_B^r = 1 - \rho_B(Z^r). \tag{44}$$

Then

$$\mathcal{R}_{\text{dec}}^r(Z^r) = \varepsilon_A^r u_A^r + \varepsilon_B^r u_B^r. \tag{45}$$

By defining

$$\varepsilon_{\text{eff}}^r = \frac{\varepsilon_A^r u_A^r + \varepsilon_B^r u_B^r}{u_A^r + u_B^r}, \tag{46}$$

we obtain

$$\mathcal{R}^r_{\text{dec}}(Z^r) = \varepsilon^r_{\text{eff}}(u^r_A + u^r_B) = \varepsilon^r_{\text{eff}}\mathcal{U}^{r,\star}. \tag{47}$$

Therefore,

$$\frac{\partial \mathcal{R}^r_{\text{dec}}}{\partial \varepsilon^r_{\text{eff}}} = \mathcal{U}^{r,\star} = u^r_A + u^r_B. \tag{48}$$

This shows that, for the same effective contribution error level, regions with larger modality-specific useful information suffer larger information regret.

### D.2. Monotonicity of Evidence Insufficiency

The evidence-insufficiency coefficient is defined as

$$U^r = \frac{2\lambda}{e^r_A + e^r_B + 2\lambda}. \tag{49}$$

Let $s^r = e^r_A + e^r_B$. Then

$$U^r = \frac{2\lambda}{s^r + 2\lambda}. \tag{50}$$

Taking the derivative with respect to $s^r$, we have

$$\frac{\partial U^r}{\partial s^r} = -\frac{2\lambda}{(s^r + 2\lambda)^2} < 0. \tag{51}$$

Therefore, $U^r$ monotonically decreases as the total observational evidence increases. This means that sufficient evidence leads to lower uncertainty and allows stronger modality allocation, while insufficient evidence produces higher uncertainty and pulls the decision toward conservative mixing.

Moreover, from

$$\bar{W}^r_B = (1 - U^r)W^r_B + 0.5U^r, \tag{52}$$

we obtain

$$\bar{W}^r_B - \frac{1}{2} = (1 - U^r)\left(W^r_B - \frac{1}{2}\right). \tag{53}$$

Thus, uncertainty modulation does not change the evidence-induced decision direction, but attenuates the decision magnitude when evidence is insufficient.

### D.3. Limitation of Implicit Deterministic Policies

We analyze why implicit deterministic fusion policies can be unreliable in discrepancy-sensitive regions. Let $O^r_{\text{imp}}$ denote the implicit decision state used by a conventional fusion model in region $r$, which may include encoded features, attention responses, gating activations, semantic priors, or other hidden representations. A deterministic implicit fusion policy produces a local contribution weight as

$$\widetilde{W}^r_B = g(O^r_{\text{imp}}), \qquad \widetilde{W}^r_B \in [0, 1]. \tag{54}$$

The corresponding allocation confidence is measured by its deviation from conservative mixing:

$$m^r_{\text{imp}} = \left|\widetilde{W}^r_B - \frac{1}{2}\right|. \tag{55}$$

A larger $m^r_{\text{imp}}$ indicates a more aggressive modality preference.

Let the ideal contribution ratio of modality $B$ be

$$w^{r,\star}_B = \frac{u^r_B}{u^r_A + u^r_B}. \tag{56}$$

The ideal contribution direction is

$$D^{r,\star} = \mathbb{I}\left[w_B^{r,\star} > \frac{1}{2}\right] = \mathbb{I}\left[u_B^r > u_A^r\right]. \tag{57}$$

Similarly, the implicit policy induces a direction

$$\widetilde{D}^r = \mathbb{I}\left[\widetilde{W}_B^r > \frac{1}{2}\right]. \tag{58}$$

The direction error probability is

$$P_{\text{imp}}^r = \Pr\left(\widetilde{D}^r \neq D^{r,\star} \mid O_{\text{imp}}^r\right). \tag{59}$$

In discrepancy regions, the implicit state $O_{\text{imp}}^r$ may entangle complementary information, noise, degradation, over-exposure, and response bias. If these decision-relevant factors are not explicitly separated, the ideal direction $D^{r,\star}$ can remain uncertain under $O_{\text{imp}}^r$. For binary direction decisions, the conditional entropy and error probability satisfy the Fano-type lower bound:

$$P_{\text{imp}}^r \geq h^{-1}\left(H(D^{r,\star} \mid O_{\text{imp}}^r)\right), \tag{60}$$

where $h(\cdot)$ is the binary entropy function. Thus, if $H(D^{r,\star} \mid O_{\text{imp}}^r)$ is large, the direction error probability of an implicit deterministic policy cannot be arbitrarily small.

We further connect this error probability to information regret. Assume that the decision-induced regret is proportional to the effective contribution error:

$$\mathcal{R}_{\text{dec}}^r = \varepsilon_{\text{eff}}^r \mathcal{U}^{r,\star}, \qquad \mathcal{U}^{r,\star} = u_A^r + u_B^r. \tag{61}$$

If the implicit policy makes a wrong directional decision with confidence at least $\tau > 0$, *i.e.*,

$$\widetilde{D}^r \neq D^{r,\star}, \qquad m_{\text{imp}}^r = \left|\widetilde{W}_B^r - \frac{1}{2}\right| \geq \tau, \tag{62}$$

then, under a mild monotonicity assumption that a larger wrong contribution deviation induces no smaller useful-information suppression, there exists a constant $c > 0$ such that

$$\varepsilon_{\text{eff}}^r \geq c\,\tau. \tag{63}$$

Therefore, the conditional expected regret of an implicit deterministic policy satisfies

$$\mathbb{E}\left[\mathcal{R}_{\text{dec}}^r \mid O_{\text{imp}}^r\right] \geq c\,\tau\,P_{\text{imp}}^r \mathcal{U}^{r,\star}. \tag{64}$$

Using the Fano-type lower bound, we further obtain

$$\mathbb{E}\left[\mathcal{R}_{\text{dec}}^r \mid O_{\text{imp}}^r\right] \geq c\,\tau\,h^{-1}\left(H(D^{r,\star} \mid O_{\text{imp}}^r)\right)\mathcal{U}^{r,\star}. \tag{65}$$

This shows that, when decision-relevant factors remain entangled in the implicit state and the policy makes over-confident allocations, the expected regret is lower bounded by decision ambiguity, over-confidence, and local useful information.

