# OpenReview forum: "Difference-Aware Decision Learning for Multimodal Image Fusion"
_ICML.cc/2026/Conference — ICML 2026 regular_

### Official Review · Reviewer_FLFv · 2026-03-09

**Soundness:** 3
**Presentation:** 3
**Significance:** 3
**Originality:** 3
**Overall Recommendation:** 5
**Confidence:** 5

**Summary:**

The paper presents a probabilistic decision framework for infrared and visible image fusion that stabilizes modality selection under large cross-modal discrepancies. The key contributions include a Prior Decision and Posterior Risk Minimization formulation to cast fusion as a closed-loop decision process, a Modality Difference-Aware (MDA) module to localize spatially salient discrepancies, and a Condition-Driven Gating (CDG) mechanism. It constructs frequency-domain evidence and maps it to per-pixel fusion weights via a symmetric Beta prior with uncertainty-aware modulation.

**Compliance With Llm Reviewing Policy:**

Affirmed.

**Final Justification:**

The rebuttal has addressed my questions. I will maintain my rating.

**Key Questions For Authors:**

How does the author prove the consistency of the risk definition in formula (1) and formulas (8)~(10)? Why are spectrum energy, complementary terms, and entropy/reliability all necessary core decision conditions, instead of being ablated separately?

**Limitations:**

The authors have not discussed the limitations or potential societal impact. Addressing issues like scalability, biases, and ethical concerns in sensitive areas would be beneficial.

**Strengths And Weaknesses:**

Strengths:
1. The paper combines a Prior Decision and Posterior Risk Minimization formulation with a Condition-Driven Gating mechanism for probabilistic fusion.
2. The paper addresses an important problem in infrared and visible image fusion: unstable modality selection under large cross-modal discrepancies.
3. The paper includes ablation experiments, efficiency analysis, cross-dataset evaluation, and downstream detection experiments. The overall experimental design is reasonably comprehensive.
Weaknesses:
1.The modulation result of branch A in Formula 27 is calculated using B. Why not use A to calculate it?
2.The paper mentions that existing methods have already used text or other auxiliary inputs, so what exactly does it mean when it says that "existing methods still only observe the source image"?
3.In the Formalization, Eq. (1) writes the risk as r_{\omega,\theta}(X), while Eq. (8)–(10) changes it to r_{\omega,\theta}(Z;X) and uses it for posterior inference. Please clarify whether these two represent the same risk function and its precise definition.
4.Power-spectrum energy, complementary terms, and spectral entropy / decision reliability are defined as core decision conditions. The authors are requested to provide ablation of each condition to verify their independent contribution and necessity.
5.What are the advantages of symmetric beta priors compared to standard sigmoid/softmax gating?

---

> ### Author Rebuttal · Authors · 2026-03-29
>
> Thank you very much to the reviewers for their careful reading and these five very insightful questions. These questions have helped us further clarify the key technical points and conceptual expressions in our methodology. Our responses to each question are as follows.
> 1. Thank you to the reviewer for pointing out this issue. The reviewer's question is entirely correct. According to the current writing style of the paper, the second term in Eq. (27) is incorrectly written as F^B_l again when defining the modulation result of branch A. This is actually a typo. We will correct this typo in the revised manuscript.
> 2. Thank you for the reviewer's question. We intended to express a more specific point. What we really meant was that, for the fusion allocation process itself, many existing methods still primarily rely on implicit transformations of source modal features to make decisions, without explicitly constructing a discrepancy-aware observation space specifically for decision formation. Even when some methods introduce auxiliary inputs such as text, these inputs are often used more for semantic modulation or representation enhancement than as an explicit probabilistic observation variable to parameterize region-level fusion decisions. This is the distinction we wanted to emphasize in Sec. 1. Therefore, we don't mean that all existing methods use only raw source images and have no other inputs; rather, they typically don't explicitly define fusion as: observation cue extraction → prior decision formation → posterior risk correction.
> 3. Thank you to the reviewer for raising this crucial question. Yes, both notations refer to the same underlying risk, just at different levels of explicitness: r_ω,θ​(Z;X):=ℓ_θ​(Fω​(X,Z);X). This expression clearly shows that for the same input 𝑋, different decisions 𝑍 will induce different fusion outputs, and therefore different risks. This is also the form that posterior inference must take, because posterior itself is defined on the decision variable. In fact, Appendix B has explicitly given this more accurate decision-level notation.
> 4. **Ablation of decision conditions inside CDG.** We further evaluate the independent contribution of each frequency-domain decision condition by removing one condition at a time while keeping the others unchanged.
>
> | Method | Energy | Complementary | Entropy/Reliability | EN↑ | SF↑ | AG↑ | SD↑ | SCD↑ |
> |--------|--------|---------------|---------------------|-----|-----|-----|-----|------|
> | Full model | ✓ | ✓ | ✓ | 7.0327 | 12.6222 | 4.4290 | 50.9291 | 1.6649 |
> | w/o Energy | ✗ | ✓ | ✓ | 6.7225 | 12.1675 | 4.3785 | 48.7267 | 1.6426 |
> | w/o Complementary | ✓ | ✗ | ✓ | 6.8663 | 12.2389 | 4.2673 | 47.3873 | 1.6335 |
> | w/o Entropy/Reliability | ✓ | ✓ | ✗ | 6.9895 | 12.5689 | 4.4108 | 49.5784 | 1.6502 |
>
> From the table, removing any single condition degrades performance, indicating that the three conditions play complementary roles in decision formation: energy captures signal strength, complementary terms characterize cross-modal exclusive frequency components, and entropy/reliability suppresses unreliable decisions under noisy or ambiguous observations.
> 5. Thank you to the reviewer for raising this issue. Its core advantage lies in the fact that symmetric beta-prior-style gating provides an explicit uncertainty-aware conservative bias, which standard sigmoid/softmax gating typically does not naturally provide in the same interpretable way. More specifically:
>
> (i) In our symmetric form, the gate naturally reverts to around 0.5 when evidence is weak. This provides a clear neutral allocation point and avoids artificial bias towards a particular modality.
>
> (ii) The parameter φ has explicit interpretability; it is equivalent to prior evidence (pseudo-count) and plays a stabilizing role in decisions when observational evidence is insufficient. In contrast, while sigmoid/softmax can also normalize scores, it typically does not explicitly express "how much conservative mixing should be reverted to under low confidence conditions."
>
> (iii) Since this paper formalizes fusion as a prior-to-posterior probabilistic decision process overall, symmetric Beta-style smoothing is more consistent in interpretation than the purely heuristic sigmoid/softmax gate.

---

> > ### Author Rebuttal · Reviewer_FLFv · 2026-04-03
> >
> > Thanks to the authors for their responses. The rebuttal has addressed my questions. I will maintain my rating.

---

### Official Review · Reviewer_fuoT · 2026-03-09

**Soundness:** 4
**Presentation:** 3
**Significance:** 4
**Originality:** 4
**Overall Recommendation:** 5
**Confidence:** 5

**Summary:**

This paper treats multimodal image fusion as a PD-PRM system. Based on this, a lightweight fusion network based on difference perception is proposed, and experiments show that the proposed method achieves state-of-the-art performance.

**Compliance With Llm Reviewing Policy:**

Affirmed.

**Final Justification:**

The authors have solved all my concerns.

**Key Questions For Authors:**

Treating multimodal image fusion as a probabilistic decision-making system is novel, but the authors need to provide detailed and comprehensive explanations of many concepts during the formalization process. The entire formalization process lacks some mathematical rigor to a certain extent.

**Limitations:**

The author does not discuss the origins of the ideas presented in this paper. As this is the first time these ideas have been proposed, the paper lacks a vision for their future development.

**Strengths And Weaknesses:**

Strengths:
1. The method proposed in this paper is lightweight while ensuring performance, and the model has certain engineering application prospects.
2. Treating multimodal image fusion as a PD-PRM system is a novel approach. Furthermore, the authors have included a reasonable formalization process.
3. The difference-aware strategy proposed by the authors turns the inherent shortcomings of multimodal image fusion tasks into a key area for performance improvement, which is a very clever idea.
4. The module proposed in this paper is innovative and can be adapted to other algorithms.

Weaknesses:
1. In the probabilistic decision-making system proposed by the author, how should "probability" and "decision" be understood in multimodal fusion?
2. In Section 2, the authors first define the fusion process as $Y = F_ω(X, Z)$ and introduce the risk function $R(Y, Z; X)$. However, in the subsequent process, they state that $r_{ω,θ}(X):=l_θ(F_ω (X, Z); X)$. So, is the optimization object for minimizing the posterior risk the decision Z or the network parameter ω?
3. This article mentions several concepts: observation, prior decision-making, and risk minimization. However, the author has not clearly explained the logical connection between these three concepts.
4. Formula 27 contains a clear typo: $F ̂_A^l=F_B^l⊙(1-W_B^l)$ should be changed to $\hat{F}_A^l=F_A^l⊙(1-W_B^l)$.
5. In some visualization experiments, GIFNet appears to perform better than IDEAL, but the authors do not explain the specific reasons in the main text.

---

> ### Author Rebuttal · Authors · 2026-03-29
>
> We are very grateful to the reviewers for their careful reading and for pointing out these key issues. We agree that the explanations of several core concepts in the current manuscript are not clear enough, and some expressions and presentation issues also need to be revised. Our responses are detailed below.
> 1. Thank you to the reviewers for raising this fundamental question. In this paper, "decision" is not a semantic classification decision in the usual sense, nor is it a discrete symbol selection. More accurately, the "decision" we refer to is the internal information allocation variable in the fusion process, which determines how information from different modalities should be preserved, suppressed, or weighted in each region. Therefore, in Sec. 2, we use 𝑍 to uniformly represent internal control variables such as attention, gating, modulation, and selection. Correspondingly, "probability" describes the uncertainty of these decision variables. The prior refers to the a priori belief about possible fusion allocation methods formed based on observation cues before posterior correction; the posterior is the result obtained after balancing between "low risk" and "close to the prior." In this sense, our probabilistic formulation does not mean that the fusion output itself is a random observation, but rather that the decision process governing fusion can be modeled as an uncertainty-aware inference of the regional allocation policy.
> 2. Thank you to the reviewer for raising this very important question. Our intention is that, at the decision-inference level, when 𝐹_𝜔 and ℓ_𝜃 are fixed, posterior risk minimization optimizes the distribution of the decision variable 𝑍, i.e., the solution for 𝑞(⋅∣𝑋) in Eq. (8)/(36). At this level, the optimization object is the decision policy, not the backbone parameter 𝜔. At the network-training level, the network parameter 𝜔 is learned through empirical risk minimization, corresponding to Eq. (3)–(7). At this level, 𝜔 parameterizes the fusion operator 𝐹_𝜔, and the network gradually adapts through training, thereby reducing the overall risk of the decision-conditioned fusion process. Therefore, to put it simply that posterior risk minimization is relative to 𝑍, while standard network training is relative to 𝜔.
> 3. Thank you to the reviewer for pointing this out. We agree that this logical chain needs to be written more clearly in the main text. Our intention is as follows:
>
> （i）Observation (Π=Ψ(X)): First, observation cues are extracted from the multimodal input, such as spatial differences and frequency domain discrepancy statistics. These cues themselves are not decisions, but rather provide evidence for modality discrepancy, complementarity, and reliability in the local region.
>
> （ii）Prior decision-making (p_α(Z|Π)): Based on these observations, the model forms prior beliefs about the region-level fusion decision. In our implementation, this step is specifically handled by the MDA and CDG modules, which transform discrepancy-related observations into gating/allocation weights.
>
> (iii)Posterior risk minimization (q_α(Z|X)):Then, the prior decision is modified by minimizing the expected risk while maintaining closeness to the prior. This step ensures that the final decision policy is not determined solely by the observations, but is also constrained and corrected by the downstream fusion objective.
> 4. Thank you very much to the reviewer for pointing out this issue. The reviewer is absolutely correct. This is indeed a typo in Eq. (27). The correct form should be: 𝐹^𝐵_𝑙=𝐹^𝐵_𝑙⊙𝑊^𝐵_𝑙,𝐹^𝐴_𝑙=𝐹^𝐴_𝑙⊙(1−𝑊^𝐵_𝑙).F^B_l=F^B_l⊙W^B_l,F^A_l=F^A_l⊙(1−W^B_l).
> 5. Thank you to the reviewer for pointing this out. This is a very valid point. Our understanding is that methods like GIFNet sometimes produce sharper, more prominent visual effects in local regions due to their more aggressive enhancement of high-frequency components. In some qualitative examples, this effect makes them appear to have stronger local contrast or saliency. However, this does not necessarily equate to more balanced fusion quality, as excessive high-frequency enhancement can also lead to over-sharpening, modality bias, or artifacts. This understanding is also consistent with our quantitative results: on multiple datasets, our method outperforms metrics reflecting overall balance, such as EN/AG/SD, while remaining competitive on other metrics.

---

> > ### Author Rebuttal · Reviewer_fuoT · 2026-04-03
> >
> > The authors have solved my concerns, so I will raise my score to 5.

---

### Official Review · Reviewer_2hQP · 2026-03-10

**Soundness:** 3
**Presentation:** 4
**Significance:** 4
**Originality:** 4
**Overall Recommendation:** 5
**Confidence:** 4

**Summary:**

This work addresses the issue of decision uncertainty in multimodal image fusion arising from large cross-modal discrepancies and local conflicts. Furthermore, it proposes a novel fusion paradigm named Difference-aware Decision Learning for Multimodal Image Fusion. The image fusion task is formally cast as a closed-loop probabilistic decision system incorporating Prior Decision and Posterior Risk Minimization. The core idea of the method is to treat modal discrepancies as decision-triggering signals. Specifically, the MDA module is designed to extract spatial-domain differences, while a CDG module analyzes spectral energy, dispersion, and entropy in the frequency domain to guide decision generation. Furthermore, the model employs a Beta-prior-based to map gating weights and establishes an uncertainty modulation mechanism to enhance robustness.

**Compliance With Llm Reviewing Policy:**

Affirmed.

**Final Justification:**

The response well addressed my concerns.

**Key Questions For Authors:**

(1) The manuscript introduces a decision gating mapping based on the Beta distribution, but the parameter selection strategy and its task dependence are not clearly explained.

(2) In the CDG module, feature maps are partitioned into patches for frequency analysis. How does the patch size influence the reliability of frequency statistics and the final fusion performance?

(3) The threshold (0.5) used to suppress updates under insufficient observations lacks justification. What is the rationale behind this choice?

(4) The risk function involves several hyperparameters (e.g., intensity and gradient weights). How sensitive are these parameters across different fusion tasks?

**Limitations:**

(1) The method introduces multiple manually defined parameters, which may require tuning across different tasks.

(2) The effectiveness of frequency-domain discrepancy statistics across heterogeneous modalities remains uncertain.

(3) The robustness under severe noise conditions in both modalities is not fully discussed.

(4) The distinct contributions of spatial and frequency discrepancies are not sufficiently analyzed.

**Strengths And Weaknesses:**

Strengths:
(1) This manuscript redefines image fusion as a closed-loop probabilistic decision system. It mathematically demonstrates that minimizing posterior risk can continuously refine prior decisions.

(2) Unlike conventional methods that focus solely on spatial intensity. This work leverages discrepancy features in both the spatial domain and the frequency domain as the basis for decision-making, thereby enabling more comprehensive and reliable decisions.

(3) The manuscript employs a symmetric Beta distribution to map decision conditions into gating weights. Compared to black-box feature stacking, this approach renders the contribution of each modality more explicit and interpretable.

(4) This manuscript proposed an uncertainty modulation mechanism, and it enables the system to automatically revert to a conservative fusion strategy under insufficient or unreliable observations. Furthermore, it effectively preventing the propagation of artifacts caused by erroneous decisions.

Weaknesses:
(1)The shape of the Beta distribution is governed by its parameters. What is the specific parameter selection strategy when mapping observation conditions to gating weights? Has this been adjusted for different tasks?

(2)In the CDG module, feature maps are partitioned into patches. How does the patch size specifically affect the accuracy of frequency information extraction and the resulting fusion quality?

(3)The motivation behind Figure 1 is not clearly conveyed. It is recommended to refine both the figure and its accompanying description for clarity.

(4)The experiments cover infrared–visible, multi-exposure, and multi-focus fusion. Given this, does the model’s frequency-domain discrepancy statistics remain representative when applied to modalities with fundamentally different imaging mechanisms?

(5)The manuscript states that updates are suppressed under “insufficient observation conditions.” In Eq. (27), this threshold is set to 0.5. Please clarify the rationale behind selecting this value.

(6)The risk function in Eq. (29) incorporates both intensity and gradient constraints. How sensitive are the hyperparameters ( κ ) and ( ρ ) across different scenarios? Do they require careful tuning for specific tasks?

(7)Spectral entropy is employed as a reliability factor. If both input modalities contain severe noise, could the model fail to identify a reliable decision, thereby leading to an overly smoothed output?

(8)In the ablation studies of the MDA and CDG modules, if only one of the modules is retained, which evaluation metric exhibits the most significant performance degradation? Does this observation reveal the respective roles of spatial discrepancies and frequency-domain discrepancies in the decision-making process?

---

> ### Author Rebuttal · Authors · 2026-03-29
>
> 1. Thank you for the reviewer's questions. In our implementation, instead of using asymmetric Beta distribution family parameters that are individually tuned for different tasks, we use a symmetric Beta-prior style smoothing approach to map observation evidence to gating weights. According to Appendix A, we uniformly set 𝜆 = 1 in all experiments. Therefore, we are currently using a cross-task shared parameter strategy, rather than tuning parameters separately for infrared-visible light, multiple exposures, and multiple focus. There are two reasons for this design: (i) to avoid artificially imposing a priori bias on a particular modality; (ii) to naturally bring the decision back to uncertainty-aware conservative mixing when the evidence is weak.
> 2. There are two reasons for this design: Thank you for the reviewer's question. Patch size effectively controls the trade-off between spectral stability and spatial locality in frequency domain cue extraction. Larger patches tend to provide more stable local spectra, thus yielding more stable energy/entropy estimates; however, the cost is reduced spatial positioning accuracy, and fine-grained regional decisions become coarser. Smaller patches retain stronger locality, making them more suitable for fine-grained regional decisions, but their spectral statistics are more susceptible to local noise disturbances, resulting in relatively weaker stability. (ii) When evidence is weak, it naturally pulls the decision back to uncertainty-aware conservative mixing.
> 3. Thank you for the reviewers' comments. Currently, Figure 1 does not clearly convey the core motivation. Our intention is to illustrate that under strong cross-modal discrepancy, many fusion models are easily influenced by a dominant mode, while our method explicitly uses discrepancy as an observation-conditioned cue to guide regional decisions. However, the current diagram design and wording, especially the expression "Correct Decision / Wrong Decision," are indeed prone to misunderstanding.
> 4. This is an excellent question. Our view is that the frequency-domain discrepancy statistics used in this paper are not tied to a specific imaging mechanism, but rather describe a more general class of local signal organization properties: information intensity (energy), complementary frequency content, and uncertainty/noise tendency (entropy). Therefore, we consider them as task-agnostic observation cues, rather than some modality-specific physical quantity. This is also consistent with the formalization in Appendix B.7/B.8, whereby the PD–PRM framework itself is cue-agnostic, allowing for the substitution of different cue instantiations.
> 5. Thank you for the reviewer's question. The 0.5 here should more accurately be understood as a neutral mixing point, rather than an externally imposed hard threshold. When uncertainty is high, the gating weight is pulled towards 0.5, meaning neither mode is strongly biased. This design stems from our symmetric prior: if observational evidence is insufficient to support a reliable bias, the safest choice is balanced allocation, rather than forcibly biasing towards one mode.
> 6. Thank you for the reviewer's questions. In Appendix A, we uniformly set λ=λ=10 for all experiments. That is, we used the same set of parameters for all three tasks: infrared-visible light, multiple exposure, and multiple focus, without performing individual parameter tuning for specific scenarios. Thank you for the reviewer's questions. The experimental results show that the model remains stable under this shared setting, indicating that the method does not heavily rely on fine-tuning of these two coefficients in a task-by-task manner. This is because the effects of these two losses are highly complementary and relatively standard: the intensity term primarily stabilizes global brightness/structure preservation, while the gradient term mainly helps preserve edge and local details.
> 7. Yes, this is possible in extreme cases, and it's actually consistent with our method design goals. If both modalities are severely degraded, then observation conditions shouldn't provide high-confidence evidence in the first place. In our design, higher spectral entropy reduces reliability, thereby reducing effective evidence, increasing uncertainty, and further pulling the gating decision back to conservative mixing.
> 8. Thank you for the reviewer's questions. As Table 4 shows, when only a portion of the discrepancy module is retained, the performance degradation does not occur uniformly across all metrics, which indeed reflects the different roles of the two approaches. From a more conceptual perspective: MDA focuses more on answering where discrepancy matters spatially; CDG focuses more on answering how modality preference should be determined from structured frequency-domain evidence.

---

> > ### Author Rebuttal · Reviewer_2hQP · 2026-04-07
> >
> > Thanks for the response. Most of my concerns are addressed, and I tend to maintain my positive score.

---

### Official Review · Reviewer_yUyr · 2026-03-10

**Soundness:** 3
**Presentation:** 3
**Significance:** 3
**Originality:** 3
**Overall Recommendation:** 4
**Confidence:** 5

**Summary:**

The authors propose IDEAL, a difference-aware multimodal fusion framework that treats cross-modal differences as decision signals and learns modality contribution policies under local conditions. By leveraging difference-attention, frequency-domain features, and uncertainty-aware gating, IDEAL adaptively determines modality contributions and improves fusion robustness. Experiments demonstrate stable and competitive performance across multiple benchmarks.

**Compliance With Llm Reviewing Policy:**

Affirmed.

**Final Justification:**

Most of my concerns are addressed. I will maintain my positive score.

**Key Questions For Authors:**

Please refer to Weaknesses.

**Limitations:**

The author did not discuss its limitations.

**Strengths And Weaknesses:**

# Strengths
1. The authors provide a novel view for the image fusion task, and the effectiveness of this method has been verified through extensive experiments.
2. It also provides a solid theoretical proof for the closed-loop probabilistic decision system, which is valuable for image fusion research.
3. The paper is well-organized, and the experimental design is generally rigorous.

# Weaknesses
1. The paper claims that relying solely on a convolutional network for prior decision-making may restrict available observation cues and lead to overly simplistic decision criteria, potentially causing low-contrast or small-scale targets to be overlooked. However, this claim would benefit from theoretical justification or empirical evidence or references. To my knowledge, CNN-based methods can also effectively capture small-scale target features in many scenarios.
2. In Figure 1, there appear to be some minor inconsistencies. For example, the label “Correct Decision” seems to refer to the method proposed in the manuscript. However, according to the paper, the decision is made at the network stage. If so, how can we determine whether the decision is correct or incorrect before the input enters the network?
3. The author states that “as the model grows, many implicit intermediate decisions become increasingly difficult to control.” What is the basis for this claim? Is there empirical or theoretical evidence supporting it?
4. The proposed method uses power-spectrum energy, complementary spectra, and spectral-entropy reliability as observation signals. It would be interesting to understand whether incorporating additional observation cues could further improve performance, and how sensitive the method is to the choice of these specific observations.

---

> ### Author Rebuttal · Authors · 2026-03-29
>
> We are very grateful to the reviewers for their careful reading and constructive comments. We agree that several statements in the current manuscript could be further strengthened and clarified. We will revise the paper accordingly.
> 1. We thank the reviewers for pointing this out. We agree that the current statement is indeed too strong a generalization. Our intention was not to say that CNNs are inherently incapable of capturing small-scale or low-contrast objects. What we really meant was a narrower proposition: when prior decisions are driven solely by implicit spatial features of the source image, the observation cues upon which the decisions rely are often insufficiently explicit in regions with strong cross-modal differences or weak local saliency. In such cases, the model can only indirectly infer discrepancy, reliability, and complementarity through implicit feature transformations, which makes the decision criteria even less explicit and harder to control. In response to reviewers' concerns, we will revise the main text to make the statement more precise and avoid generalization. Specifically, we will change the original somewhat absolute capability assessment to a statement about decision conditioning: that is, when discrepancy/reliability cues are not explicitly exposed, CNN-only prior decisions may not be reliable in some complex regions. At the same time, we will also add empirical support: ablation experiments show that when the discrepancy-aware decision component is removed, the EN/AG/SD metrics on multiple datasets consistently decrease, indicating that explicit cue construction does indeed improve decision quality compared to relying solely on the backbone itself.
> 2. We thank the reviewer for pointing out this issue. We agree that the current representation in Figure 1 may be misleading. The reviewer's understanding is correct: it is impossible to determine whether a decision is "correct" or "wrong" before the input has entered the network. Our original intention was to express a retrospective qualitative interpretation, namely, whether the final fusion result preserved more reasonable modal information in cross-modal conflict regions.
> 3. Thank you to the reviewer for pointing this out. We agree that this statement needs clearer evidence. Our intention was not simply to say "the larger the model, the worse it is," but rather that if we only increase the model capacity without explicitly defining the internal decision variables, the internal fusion decision-making process may become stronger, but it will not automatically become more interpretable or less controllable. In our framework, fusion essentially involves regional-level allocation, suppression, and preservation decisions; if these decisions are only implicitly carried by hidden activations without explicit cue-conditioned structures, it becomes more difficult to diagnose "which observation triggered which allocation behavior." This is why we explicitly introduced cue variables and decision variables in the PD–PRM formalization. To make this statement more rigorous, we will rewrite the original sentence as something like the following: "Scaling network capacity alone may improve representation power, but when fusion allocation is governed only by implicit hidden activations, the resulting intermediate decision process becomes less interpretable and less directly controllable."
> 4. This is a good suggestion. Our proposed framework is not bound to the frequency domain cues used in this paper. We chose power-spectrum energy, complementary spectra, and spectral-entropy reliability because they correspond to discrepancy magnitude, complementary content, and uncertainty/reliability, respectively, offering strong interpretability while having low computational cost and simple implementation. However, from the formal perspective of PD–PRM, the framework itself is cue-agnostic. As explained in the appendix, the entire framework does not depend on any particular cue choice; replacing or adding other cues does not change its mathematical structure. We will explicitly add this point in the revised version, and explain that other cues, such as local contrast, gradient consistency, cross-modal structural correlation, or task-informed saliency, which may further improve performance, especially in more specific task scenarios. Regarding sensitivity, our current experimental results show that the existing cue set already has good effectiveness and stability, as the full model consistently outperforms all ablation versions on all three datasets.

---

> > ### Author Rebuttal · Reviewer_yUyr · 2026-04-02
> >
> > Thanks to the authors’ detailed responses. I will keep my positive score. Please revise the paper accordingly to reflect the points addressed in the rebuttal.

---

### Decision · Program_Chairs · 2026-04-30

**Decision:**

Accept (regular)

**Comment:**

All four reviewers viewed the proposed method positively and generally recognized the strength of the experimental evaluation. The main concerns raised during discussion were mostly related to clarity of presentation, wording, parameter selection, the scope of the ablation study, and limitations.

Overall, all the reviewers indicated that their major questions had been adequately or largely resolved while maintaining an overall positive assessment. Based on the reviews and rebuttal discussion, I am inclined to support acceptance of this manuscript.